# Calculation of Francium Hyperfine Anomaly

**Elena A. Konovalova** [1,*] , **Yuriy A. Demidov** [1,2] , **Mikhail G. Kozlov** [1,2,*]
and **Anatoly E. Barzakh** [1]

[1]   Petersburg Nuclear Physics Institute of NRC "Kurchatov Institute", Gatchina,
     Leningrad District 188300, Russia; iurii.demidov@gmail.com (Y.A.D.); barzakh@mail.ru (A.E.B.)
[2]   Physics Department, Saint Petersburg Electrotechnical University "LETI", Prof. Popov Str. 5,
     Saint Petersburg 197376, Russia
*   Correspondence: lenaakonovalova@gmail.com (E.A.K.); mihailgkozlov@gmail.com (M.G.K.)

**Abstract:** The Dirac–Hartree–Fock plus many-body perturbation theory (DHF + MBPT) method has been used to calculate hyperfine structure constants for Fr. Calculated hyperfine structure anomaly for hydrogen-like ion is in good agreement with analytical expressions. It has been shown that the ratio of the anomalies for $s$ and $p_{1/2}$ states is weakly dependent on the principal quantum number. Finally, we estimate Bohr–Weisskopf corrections for several Fr isotopes. Our results may be used to improve experimental accuracy for the nuclear $g$ factors of short-lived isotopes.

**Keywords:** hyperfine anomaly; hyperfine structure; nuclear charge distribution; Bohr–Weisskopf effect

## 1. Introduction

The hyperfine structure constants (HFS) and isotope shifts are highly sensitive to the changes of charge and magnetization distributions inside the nucleus. The precision achieved in measurements of these parameters coupled with advances in atomic theory enable new atomic physics based tests of nuclear models. Thus, the HFS measurements can serve as very useful tool, for example, for the understanding of the shape coexistence phenomena in atomic nuclei [1].

The ratio of magnetic hyperfine constants $A$ for different isotopes is usually assumed to be equal to the ratio of their nuclear $g$ factors $g_I = \frac{\mu}{\mu_N I}$, where $\mu$ and $I$ are magnetic moment and spin of the nucleus, $\mu_N$ is nuclear magneton. However, this is true only for the point-like nucleus. For the finite nucleus, one should take into account: (i) the distribution of the magnetization inside the nucleus; and (ii) the dependence of the electron wave function on the nuclear charge radius. Former correction is called magnetic, or Bohr–Weisskopf (BW) correction [2] and the latter one is called charge, or Breit–Rosenthal (BR) correction [3,4]. These corrections break proportionality between magnetic hyperfine constants and nuclear $g$ factors. This phenomenon is called the hyperfine anomaly (HFA) [2]. Magnetic HFA gives the unique opportunity to trace the change in the magnetization distribution in the nucleus. However, the absence of advanced atomic calculation prevents extracting the nuclear parameters from the experimental data. On the other hand, the HFA correction is usually small (less than 1%) [5] and only recently the measurements for the short-lived nuclei reached this level of accuracy. Thus, the development of the new methods of atomic calculations of the hyperfine constants accounting for the HFA becomes relevant and timely [6]. Below, we discuss how to calculate HFA for many-electron atoms with available atomic package [7], which is based on the original Dirac–Hartree–Fock code [8]. This package has often been used to calculate different atomic properties including HFS constants of Tl [9–11] and Pb [12].

We study francium atom, because for its isotopic chain there are comprehensive experimental data [13–18] and many theoretical calculations [19–22]. In particular, changes of the nuclear charge radii in the Fr isotopic series were calculated from the isotope shift measurements [23,24].

## 2. Theory and Methods

It is generally accepted that the observed HFS constant $A$ can be written in the following form (see, e.g., Reference [25]):

$$A = g_I \mathcal{A}_0 (1 - \delta)(1 - \epsilon). \tag{1}$$

Here, $g_I$ is the nuclear $g$ factor, $g_I \mathcal{A}_0$ is the HFS constant for the point-like nucleus, and $\delta$ and $\epsilon$ are the nuclear charge distribution (BR) and magnetization distribution (BW) corrections respectively. $\mathcal{A}_0$ is independent of the nuclear $g$ factor. In the case of hydrogen-like ions, the expression for $\mathcal{A}_0$ was obtained in the analytical form by Shabaev [26]:

$$\mathcal{A}_0 = \frac{\alpha(\alpha Z)^3}{j(j+1)} \frac{m}{m_p} \frac{\varkappa(2\varkappa(\gamma + n_r) - N)}{N^4 \gamma(4\gamma^2 - 1)} mc^2. \tag{2}$$

Here, $\alpha$ is the fine-structure constant, $Z$ is the nuclear charge, $m$ and $m_p$ are electron and proton masses, $j$ is the total electron angular momentum, $\varkappa$ is the relativistic quantum number, $N = \sqrt{n_r^2 + 2n_r\gamma + \varkappa^2}$, $n_r$ is the radial quantum number, and $\gamma = \sqrt{\varkappa^2 - (\alpha Z)^2}$. We use refined model of the homogeneously charged and magnetized ball of the radius $R = \left(\frac{5}{3}\langle r^2 \rangle\right)^{1/2}$. Extended nuclear magnetization is formed by the spin polarization of nucleons and by the orbital motion of protons. The charge density inside the nucleus is relatively stable for different isotopes [27], whereas the nuclear magnetization strongly depends on the spin and configuration of each isotope. Following Refs. [28,29], we introduce the nuclear factor $d_{\text{nuc}}$ for parameterization of these nuclear effects. Then, the BR and BW corrections $\delta$ and $\epsilon$ for a given $Z$ and electron state can be written as [11]:

$$\delta(R) = b_N R^{2\gamma - 1}, \qquad \epsilon(R, d_{\text{nuc}}) = b_M d_{\text{nuc}} R^{2\gamma - 1}, \tag{3}$$

where $b_N$ and $b_M$ are factors, which are independent of the nuclear radius and structure. It follows from Equations (1) and (3) that, if we calculate the HFS constant for different $R$ and $d_{\text{nuc}}$, we should get in the first order in $\delta$ and $\epsilon$ the following dependence on the nuclear radius:

$$A(g_I, d_{\text{nuc}}, R) = g_I \mathcal{A}_0 \left(1 - (b_N + b_M d_{\text{nuc}}) R^{2\gamma - 1}\right). \tag{4}$$

Within the point-like magnetic dipole approximation $d_{\text{nuc}} = 0$ and the Bohr–Weisskopf correction, $\epsilon$ is equal to zero. Then, assuming that $g_I = 1$, one can fit the HFS constant by the function:

$$A(1, 0, R) = \mathcal{A}_0 \left(1 - b_N R^{2\gamma - 1}\right). \tag{5}$$

On the other hand, for $d_{\text{nuc}} = 1$. one obtains:

$$A(1, 1, R) = \mathcal{A}_0 \left(1 - (b_N + b_M) R^{2\gamma - 1}\right). \tag{6}$$

Let us compare HFS constants for two isotopes with nuclear $g$ factors $g_I^{(1)}$ and $g_I^{(2)}$, slightly different nuclear radii $R^{(1,2)} = R \pm \mathfrak{r}$, and nuclear factors $d_{\text{nuc}}^{(1)} = d_{\text{nuc}}^{(2)} = 0$:

$$\frac{A(g_I^{(1)}, 0, R + \mathfrak{r})}{A(g_I^{(2)}, 0, R - \mathfrak{r})} \approx 1 + 2\mathfrak{r} \frac{\partial A(g_I^{(1)}, 0, R)/\partial R}{A(g_I^{(2)}, 0, R)}. \tag{7}$$

Then, the part of the HFS anomaly related to the change of the nuclear charge distribution $^1\Delta_{\text{BR}}^2(R)$ is:

$$^1\Delta_{\text{BR}}^2(R,\mathfrak{r}) \equiv \frac{g_I^{(2)} A(g_I^{(1)}, 0, R+\mathfrak{r})}{g_I^{(1)} A(g_I^{(2)}, 0, R-\mathfrak{r})} - 1 \approx 2(2\gamma-1)b_N R^{2\gamma-2}\mathfrak{r}. \tag{8}$$

In this work, we calculate the magnetic hyperfine constants and HFS anomalies for low-lying states of Fr atom within the Dirac–Hartree–Fock (DHF) approximation and the DHF plus many-body perturbation theory (DHF + MBPT) method. In our calculations, we account for the Breit corrections and spin-polarization of the core.

## 3. Results and Discussion

### 3.1. HFS Anomaly for H-Like Francium Ion

Here, we calculate HFS constants of the $1s$, $2s$, and $2p_{1/2}$ states of Fr$^{86+}$ for the different nuclear radii $R$ and compare our results with analytical expressions from Reference [26]. Figure 1 shows the dependence of the hyperfine constant $A(1s)$ on the nuclear radius $R$ assuming that $g_I = 1$. We see very good agreement with Equations (5) and (6).

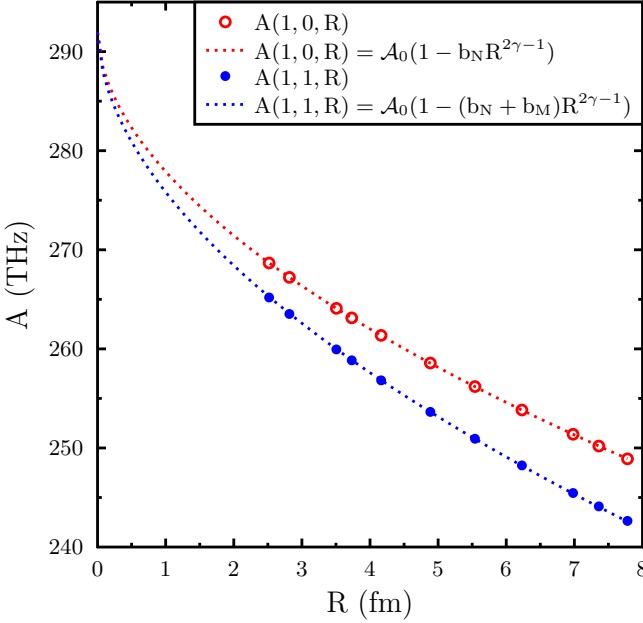

**Figure 1.** Dependence of the HFS constant $A(g_I, d_{\text{nuc}}, R)$ for the ground state of H-like Fr ion on the nuclear radius for $g_I = 1$. Dots and circles correspond to the calculated values. Dashed lines correspond to the fits by Equations (5) and (6). Dots: both BR and BW corrections are taken into account; circles: BW correction is equal to zero.

Table 1 summarizes our results for the H-like Fr ion. For all three states, we see good agreement between analytical values of $\mathcal{A}_0$ from Equation (2) and the values obtained from the fit of the calculated HFS constants for finite nuclei. According to our calculations, the ratios of the parameters $b_N$ and $b_M$ for $1s$ and $2s$ states are close to unity: $\frac{b_N(1s)}{b_N(2s)} = 0.933$ and $\frac{b_M(1s)}{b_M(2s)} = 0.933$. This is expected, as in the first approximation, inside the nucleus the wave functions of the same symmetry should be proportional to each other. Similar ratios for $2s$ and $2p_{1/2}$ states are $\frac{b_N(2s)}{b_N(2p_{1/2})} = 3.128$ and $\frac{b_M(2s)}{b_M(2p_{1/2})} = 2.961$. Again, one can expect that these ratios weakly depend on the principle quantum number.

**Table 1.** Compilation of the fitting parameters for the HFS of the H-like Fr ion. BR and BW corrections $\delta$ and $\epsilon$ for $^{210}$Fr$^{86+}$ are calculated for $R = 7.1766$ fm [30] and $d_{\rm nuc} = 1$. Parameters $\delta$ and $\epsilon$ are given in percent.

|  |  | **1s** | **2s** | **2p$_{1/2}$** |
|---|---|---|---|---|
| $\mathcal{A}_0$ (THz) | fit. | 292.0 | 49.5 | 15.2 |
|  | Reference [26] | 291.5 | 49.5 | 15.1 |
| $b_N \cdot 10^2 / {\rm fm}^{2\gamma-1}$ | fit. | 4.817 | 5.161 | 1.650 |
| $\delta(^{210}{\rm Fr})$, % | fit. | 14.11 | 15.12 | 4.83 |
| $b_M \cdot 10^2 / {\rm fm}^{2\gamma-1}$ | fit. | 0.710 | 0.761 | 0.257 |
| $\epsilon(^{210}{\rm Fr}, d_{\rm nuc} = 1)$, % | fit. | 2.08 | 2.23 | 0.75 |

*3.2. HFS Anomaly of Neutral Francium Atom*

The ground configuration of the neutral francium atom is [Rn]7$s$. If we treat francium as one-electron system with the frozen core, we can do calculations using Dirac–Hartree–Fock (DHF) method. In this case, the dependence of the HFS constants on the nuclear radius is similar to the one-electron ion.

In the DHF approximation, the HFS constant $A(7p_{3/2})/g_I = 55.6$ MHz is small and insensitive to HFA (see Table 2). At the same time, the HFS constants $A(7s)/g_I$ and $A(7p_{1/2})/g_I$ are well described by Equations (5) and (6). According to our calculations, the ratios of coefficients $b_N$ and $b_M$ for $s$ and $p_{1/2}$ waves are close to the respective ratios in H-like ion $\frac{b_N(1s)}{b_N(7s)} = 0.908$ and $\frac{b_M(1s)}{b_M(7s)} = 0.929$. This result is compatible with the assertion made in Reference [31] that the hyperfine anomaly measured for the $s$ states in Rb is weakly dependent on the principal quantum number $n = 5, 6, 7$. Ratios of the parameters $b_N$ and $b_M$ for 7$s$ and 7$p_{1/2}$ are: $\frac{b_N(7s)}{b_N(7p_{1/2})} = 2.907$ and $\frac{b_M(7s)}{b_M(7p_{1/2})} = 2.690$, while for the H-like ion we had 3.128 and 2.961, respectively.

The situation changes when we include spin-polarization of the core via random phase approximation (RPA) corrections. These corrections lead to effective mixing of different partial waves, thus the constant $A(7p_{3/2})$ acquires contributions from the $s$ and $p_{1/2}$ waves. Due to the RPA corrections, the value of the constant $A(7p_{3/2})$ is significantly changed. At the same time, this constant becomes sensitive to the distributions inside the nucleus. To account for that, we can use Equation (3) with the same $\gamma$ as for $s$ and $p_{1/2}$ states. The RPA corrections for the 7$s$ and 7$p_{1/2}$ states are smaller than for 7$p_{3/2}$, but they are also significant. Due to the RPA corrections, the ratios of the parameters $b_N$ and $b_M$ for 7$s$ and 7$p_{1/2}$ states change by $\sim 15\%$: $\frac{b_N(7s)}{b_N(7p_{1/2})} = 3.153$ and $\frac{b_M(7s)}{b_M(7p_{1/2})} = 3.073$.

Core–valence and core–core electron correlations were taken into account within the DHF + MBPT method [7]. Electron correlation corrections significantly change $\mathcal{A}_0$ values. The parameters $b_N$ and $b_M$ also change, but the ratios of these parameters for the 7$s$ and 7$p_{1/2}$ states remain stable. Without RPA corrections, these ratios are equal to: $\frac{b_N(7s)}{b_N(7p_{1/2})} = 3.033$ and $\frac{b_M(7s)}{b_M(7p_{1/2})} = 2.663$. Final ratios were obtained in the DHF+MBPT approximation with RPA and Breit corrections:

$$\frac{b_N(7s)}{b_N(7p_{1/2})} = 3.280, \qquad \frac{b_M(7s)}{b_M(7p_{1/2})} = 3.023. \tag{9}$$

According to Mårtensson-Pendrill [21], the ratio of $b_N$ parameters obtained by scaling the Breit–Rosenthal corrections for Tl is equal to 3.2 in a good agreement with our result. For the ratio of the $b_M$ parameters, she used the value 3.0 also in agreement with our results.

Information about parameters $b_N$ and $b_M$ can be extracted from the experimentally measured ratio of the HFS constants $\rho = A(7s)/A(7p_{1/2})$. This ratio can be written as a function of the nuclear radius $R$ and the nuclear factor $d_{\text{nuc}}$:

$$1 - \frac{\rho(R, d_{\text{nuc}})}{\rho_0} \approx (b_N(7s) - b_N(7p_{1/2}))R^{2\gamma-1} + d_{\text{nuc}}(b_M(7s) - b_M(7p_{1/2}))R^{2\gamma-1}, \qquad (10)$$

where $\rho_0 = \mathcal{A}_0(7s)/\mathcal{A}_0(7p_{1/2})$. Several experimentally measured values of $\rho$ for odd–odd and even–odd isotopes [13] and corresponding fits by Equation (10) are presented in Figure 2.

**Table 2.** Calculated parameters $\mathcal{A}_0$ (MHz), $b_N$ and $b_M$ (fm$^{1-2\gamma}$) for the neutral Fr atom on a different levels of approximation.

| | $\mathcal{A}_0$ | $b_N \cdot 10^2$ | $b_M \cdot 10^2$ |
|---|---|---|---|
| | $7s$ | | |
| DHF | 7894.710 | 5.3030 | 0.7646 |
| DHF + Br | 7882.694 | 5.2989 | 0.7642 |
| DHF + MBPT | 10,602.174 | 4.7502 | 0.8584 |
| DHF + MBPT+Br | 10,581.950 | 4.7013 | 0.8506 |
| DHF + RPA | 8684.144 | 5.1092 | 0.8008 |
| DHF + Br + RPA | 8682.028 | 5.1020 | 0.8007 |
| DHF + MBPT + RPA | 11,518.484 | 4.6067 | 0.8844 |
| DHF + MBPT + Br + RPA | 11,507.415 | 4.5516 | 0.8738 |
| | $7p_{1/2}$ | | |
| DHF | 746.580 | 1.8241 | 0.2842 |
| DHF + Br | 740.251 | 1.8204 | 0.2837 |
| DHF + MBPT | 1130.031 | 1.5661 | 0.3223 |
| DHF + MBPT+Br | 1120.865 | 1.5461 | 0.3160 |
| DHF + RPA | 865.034 | 1.6205 | 0.2606 |
| DHF + Br + RPA | 861.718 | 1.6223 | 0.2627 |
| DHF + MBPT + RPA | 1308.388 | 1.4018 | 0.2929 |
| DHF + MBPT + Br + RPA | 1300.950 | 1.3879 | 0.2891 |
| | $7p_{3/2}$ | | |
| DHF | 55.524 | 0.0000 | 0.0000 |
| DHF + Br | 55.153 | 0.0000 | 0.0000 |
| DHF + MBPT | 77.870 | 0.0000 | 0.0000 |
| DHF + MBPT + Br | 77.437 | 0.0000 | 0.0000 |
| DHF + RPA | 94.984 | 1.2620 | 0.2769 |
| DHF + Br + RPA | 94.721 | 1.2545 | 0.2742 |
| DHF + MBPT + RPA | 132.482 | 1.2535 | 0.2919 |
| DHF + MBPT + Br + RPA | 131.988 | 1.2382 | 0.2843 |

Even–odd Fr isotopes with neutron number $N \leq 126$ ($A \leq 213$) have spin $I = 9/2$. The magnetic moments $\mu(A, 9/2)$ for isotopes from $A = 213$ to $A = 207$ differ by only 3% [32]. Ground states of these isotopes are regarded as pure shell-model $h_{9/2}$ states. According to Reference [33], for such states, the factor $d_{\text{nuc}}$ is also constant within the same 3% limit. The value of this factor can be calculated using the simple shell-model formula: $d_{\text{nuc}} = 0.3$ (see Reference [21]). Then, the one-parameter fit with $\rho_0$ as the free parameter gives us the following relation: $\rho = 8.456\,(1 - 0.033\,R^{2\gamma-1})$, where we used our final results for $b_N$ and $b_M$ from Table 2, or $\rho = 8.404\,(1 - 0.031\,R^{2\gamma-1})$ within the two-parameter fit. Comparing these two results, we can estimate the error bars for fitting parameters to be: $\rho_0 = 8.43(3)$ and $b_N + d_{\text{nuc}}\,b_M = 0.032(1)$. Note that the theoretical value of $\rho_0$ obtained within DHF + MBPT + Br + RPA method is equal to 8.85, which is 5% larger. Taking into account the possible few percent change of the $d_{\text{nuc}}$ factor from one isotope to another and its deviation from the shell-model value, the correspondence between fitted and calculated $\rho$ values should be regarded as satisfactory.

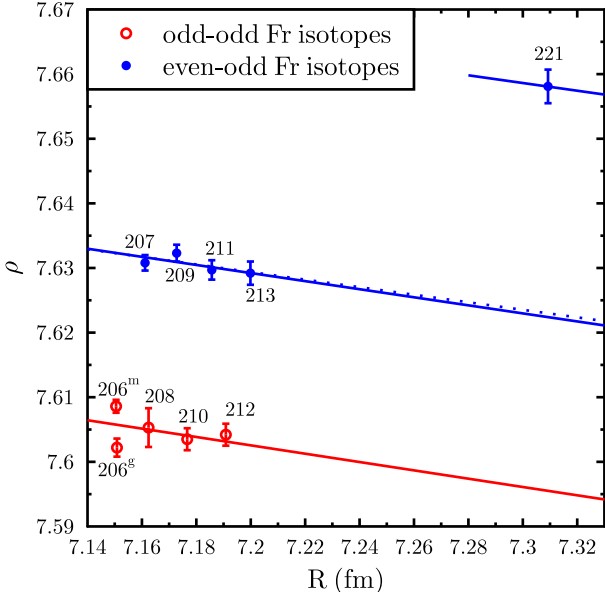

**Figure 2.** Experimentally measured ratios $\rho = A(7s)/A(7p_{1/2})$ for even–odd and odd–odd isotopes of francium [13]. The nuclear radii $R$ are taken from Reference [30]. Solid lines are the one-parameter fits by Equation (10); dashed line corresponds to the two-parameter fit. For even–odd isotopes, we use $d_{nuc} = 0.3$ [21] and parameters $\rho_0$ (one-parameter fit), or $\rho_0$ and $(b_N + d_{nuc} b_M)$ (two-parameter fit). Then, for odd–odd isotopes, we fix the value of $\rho_0$ obtained from the previous one-parameter fit and fit nuclear factor $d_{nuc}$, with the result $d_{nuc} = 0.49$. For $^{221}$Fr, the fit gives $d_{nuc} = 0.05$.

For calculation of $d_{nuc}$ factor for the odd–odd isotopes formulas from Reference [28] were used (see Equations (38) and (39) on p. 29 and Equations (41) and (42) on p. 38). Spins and configurations for these isotopes with $A = 206 - 212$ are different ($I = 5, 6, 7, 3$). Correspondingly, the factor $d_{nuc}$ can be different as well. However, in the shell model, one can show that, for all these cases, $d_{nuc} = 0.5(1)$. To check the general applicability of our approach, we suppose that the nuclear factor is the same for all considered odd–odd Fr isotopes. We fix $\rho_0$ obtained for even–odd Fr isotopes and fit nuclear factor for odd–odd ones, which gives us $d_{nuc} = 0.49$, in agreement with the shell-model estimation. The deviation of the experimental $\rho$ values for $^{206m}$Fr and $^{206g}$Fr from the fitted line (see Figure 2) is obviously connected to the structural changes in these nuclei resulting in the changes of the factor $d_{nuc}$ (see discussion in Reference [13]). For $^{221}$Fr, the fit gives $d_{nuc} = 0.05$. This result can be of a particular interest for the nuclear physics and more detailed analysis will be presented in the forthcoming paper.

The accuracy reached in our calculations of the HFS constants for neutral Fr can be estimated in comparison with available experimental and theoretical data presented in Table 3. Due to Bohr–Weisskopf correction, calculated $A(7s)$ and $A(7p_{1/2})$ constants of $^{210}$Fr are reduced approximately by 1.24% and 0.41%, respectively. Thus, within the DHF + MBPT + Br + RPA method we obtain following final values: $A(7s)/g_I = 9849.57$ MHz and $A(7p_{1/2})/g_I = 1242.94$ MHz.

**Table 3.** Calculated HFS constants for low-lying states of the neutral $^{210}$Fr. We use a point-like magnetic dipole approximation, as described by Equation (5), and assume that $R = 7.1766$ fm and $g_I = 0.733$. Then, in the final results, we add Bohr–Weisskopf correction for $d_{\text{nuc}} = 0.49$. Available experimental data and other theoretical relativistic coupled-cluster results are also presented.

| Method | $A(7s)/g_I$ (MHz) | $A(7p_{1/2})/g_I$ (MHz) | $A(7p_{3/2})/g_I$ (MHz) |
|---|---|---|---|
| DHF | 6668.56 | 706.70 | 55.52 |
| DHF + Br | 6659.37 | 700.78 | 55.15 |
| DHF + MBPT | 9127.20 | 1078.20 | 77.87 |
| DHF + MBPT + Br | 9124.92 | 1070.11 | 77.44 |
| DHF + RPA | 7491.91 | 837.58 | 95.43 |
| DHF + Br + RPA | 7496.66 | 833.22 | 94.73 |
| DHF + MBPT + RPA | 9964.42 | 1254.67 | 127.62 |
| DHF + MBPT + Br + RPA | 9973.44 | 1248.07 | 127.20 |
| FINAL (BR and BW) | 9849.57 | 1242.94 | 126.69 |
| Theory * [19] | 9927 | – | – |
| Theory [20] | 9885.24 | 1279.56 | 104.28 |
| Experiment [14,30,32] | 9856 (113) | 1296 (15) | 106.8 (13) |

\* In this study, the charge and magnetization distributions were modeled by the same Fermi distribution.

## 4. Conclusions

In this work, we use the method developed in Reference [11] to calculate the hyperfine anomaly by the analysis of the HFS constants of Fr as functions of nuclear radius. The HFA in this method can be parameterized by coefficients $b_N$ and $b_M$. We test our method by calculating HFS constants of H-like francium ion and obtain fairly good agreement with analytical expression from Reference [26]. Then, we make calculations for neutral Fr, described as a system with one valence electron. We show that the ratios of $b_N(7s)/b_N(7p_{1/2})$ and $b_M(7s)/b_M(7p_{1/2})$ are practically the same, as in H-like ion and rather stable within the DHF and DHF + MBPT approximations. However, when we include spin-polarization of the core by means of the RPA corrections, these ratios change by 10–15%.

The corrections caused by the redistribution of the magnetization inside the nucleus are estimated using experimentally measured ratio of the HFS constants $A(7s)/A(7p_{1/2})$. Estimated Bohr–Weisskopf corrections for odd–odd francium isotopes 206, 208, 210, and 212 are found to be 1.62 times larger than for even–odd isotopes 207, 209, 211, and 213. The Bohr–Weisskopf correction for $^{221}$Fr is significantly smaller than for other even–odd isotopes. This information can be used to obtain more accurate values for the nuclear $g$ factors of the short-lived isotopes of francium from the ratios of the HFS constants. The reliability of the applied method enables one to determine the nuclear factor $d_{\text{nuc}}$ which gives important nuclear-structure information and may be compared with the theoretical predictions.

**Author Contributions:** E.A.K., Y.A.D., M.G.K. and A.E.B. are equally contributed.

**Funding:** The work was supported by the Foundation for the advancement of theoretical physics "BASIS" (grant # 17-11-136).

**Acknowledgments:** Thanks are due to Vladimir Shabaev, Ilya Tupitsyn and Leonid Skripnikov for helpful discussions.

**Conflicts of Interest:** The authors declare no conflict of interest.

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
