# Peer review of "Calculation of Francium Hyperfine Anomaly"

_atoms, doi:10.3390/atoms6030039_

Round 1

Reviewer 1 Report

The paper describes calculations of the hyperfine anomaly(hfa) in Fr. 

The subject of hfa has been of minor interest since the effects were pointed out almost 70 years ago, due to a small effect, difficulties to measure hfa, few measurements and lack of suitable calculations. The paper is one in a line of papers addressing the latter. From this perspective it is well worth publishing.

However, the paper does not put hfa in a modern context when it comes to the present status of hfs, both experimentally and for calculations. A more general description would be helpful for a reader not familiar with the subject.

The presentation of the results are mainly focussed on theoretical calculations with little emphasis on experimental data. I would expect a better discussion on this.

In the tables it is not entirely clear if it is A or A/gI  that is given, especially as the unit used is MHz and not MHz/μI, could the authors be clearer on this?

Specific comments:

1. Introduction:

P.1 line 25, Why so many references to hfs calculations using the DHF-package? Maybe just the calculations in Tl and Pb should be enough.

2. Theory and Methods:

P.1 line 31 and equation 1, Do you have a reference to this equation?

p.2 line 38, You write " We use the model.." This is not entirely correct to use "use" since you change the model, it is better to use "refine".

p3 line 42, Using the factor dnuc is correct but the reader is left in confusion what it is unless they seek and read the references. a better description of it would be helpful and also discuss the values it can attain.

p3 line 43ff, You have defined BW and BR as connected to Îµ and Î´, but by using the greek letters the paper get more difficult to read. You might consider rephrasing some sentences and use BW and BR as support for better readability. For example in Figure 1.

3. Results and discussion

p5 line 76, the value of A=0.56GHz is not the same as in table 2 where A=55.6 MHz. Also without knowing that band bM is zero implies that A is independent of R this sentence can be hard to understand without a clarification. That A (3/2) is small and insensitive to hfa is expected in first approximation, something worth mentioning.

p5 line 80. The very weak dependence of n for hfa has been shown experimentally for high n's (5,6,7), to argue that this is also the case for low n is a bit risky. Results in Tl (PRA 64 032606, 2001) and (PRA 052524, 2012) are better to use.

p5 line 98 and 99, why is the ratio  bN given with one decimal  here when you use three otherwise?

p5 line 106, Since the nuclear magnetic moment of Fr is deduced from ratios of A without correcting for hfa, the changes might be larger. I would expected a discussion on this as you use this to argue on the case of hfa. I also would like a reference to the nuclear magnetic moments.

p6 line 120, How do you get this value og dnuc (ref.)

Table 3: The experimental values and errors given does not seem to correlate to the best published values. The relative error in s1/2 is on the order of 10-4 but in the table about 1%, the same for the p1/2 (compared with the article of Grossman et al.)

In general, this is a nice paper but there are a number of issues that must be addressed before the paper can be accepted. 

An improvement would be better comparisons with measurements and use of results to obtain better(?) values of the nuclear magnetic moment for Fr isotopes as has been done in J. Phys. Comm.  2 055028.

Reviewer 2 Report

The paper presents interesting new theoretical results on the hyperfine anomaly of francium. The theoretical basis and method are described clearly and transparent. The results are well presented and the length of the article is appropriate for the content.

Annoyingly the assignments of the references were missing, which strongly complicated the reading of the paper.

Round 2

Reviewer 1 Report

The manuscript has been improved and is clearer.

However, I have a few suggestions:

p1, l16 Define gas Î¼/μII.

p2, l51ff "The charge density inside..." I agree on the statement but a reference would be good.

Table1 The use of the factor 102 for bn,m is fine but not using the same factor for Î´ and Îµ makes the table somewhat difficult to read. As HFA is normally given in % should this be considered.

p5 l92ff, I still feel uneasy using only Rb as argument, but accept the authors assertion.

p6 l130. It would be helpful to indicate the specific formulas used from ref. 27. This as a service to the readers.

Even if the absolute values of the nuclear magnetic moment is depending on the value of 210, it would be nice if the relative (to 210) values could be given with errors, if and when a more accurate value is obtained by your or another group. This will increase the significance of your paper and be useful in tabulations of nuclear magnetic moments.

The work performed is impressive and innovative, improving the possibilities to finally get a better picture of the hyperfine anomaly. 

I suggest that the authors considers my suggestions as I feel these will improve the paper.

I will definitely look forward to future papers from the authors.
